# What Is Next for Public Health after COVID-19 in Italy? Adopting a Youth-Centred Care Approach in Mental Health Services

**DOI:** 10.3390/ijerph192214937

**Published:** 2022-11-13

**Authors:** Silvia Ussai, Giulio Castelpietra, Ilaria Mariani, Andrea Casale, Eduardo Missoni, Marco Pistis, Lorenzo Monasta, Benedetta Armocida

**Affiliations:** 1Clinical Pharmacology and Toxicology, University of Cagliari, 09124 Cagliari, Italy; 2Central Health Directorate, Inpatient and Outpatient Care Service, Friuli Venezia Giulia Region, 34121 Trieste, Italy; 3Department of Medicine, Surgery and Health Science, University of Trieste, 34127 Trieste, Italy; 4WHO Collaborating Centre, Institute for Maternal and Child Health—IRCCS “Burlo Garofolo”, 34137 Trieste, Italy; 5Saluteglobale.it Associazione di Promozione Sociale, 25121 Brescia, Italy; 6CERGAS—Centre for Research on Health and Social Care Management, Bocconi University, 20100 Milano, Italy; 7Department of Biomedical Sciences, Section of Neuroscience and Clinical Pharmacology, University of Cagliari, 09124 Cagliari, Italy; 8Clinical Epidemiology and Public Health Research Unit, Institute for Maternal and Child Health—IRCCS “Burlo Garofolo”, 34137 Trieste, Italy; 9Department of Cardiovascular, Endocrine-Metabolic Diseases and Aging, Istituto Superiore di Sanità, 00161 Rome, Italy

**Keywords:** mental health, COVID-19, youth, youth-centred care, resilience

## Abstract

Although endeavours to protect mental well-being during the COVID-19 pandemic were taken at national and regional levels, e.g., mental support in school, a COVID-19 emergency toll-free number for psychological support, these were sporadic conjunctural financing interventions. In this Communication, the authors conducted a systematic search for programmatic and policy documents and reports with a solid literature and policy analysis concerning the main objective, which is to analyse the appropriateness in implementing gender- and age-sensitive, integrated, youth-centred mental health services in Italy. The Italian National Action Plan for Mental Health reports a highly fragmented situation in the Child and Adolescent Neuropsychiatry services, in terms of an integrated and comprehensive regional network of services for the diagnosis, treatment, and rehabilitation of neuropsychological disorders in young people. Wide-ranging interventions, systemic actions should be implemented, funded, and included in an overall structural strengthening of the healthcare system, including those dedicated to transition support services. In this context, the National Recovery and Resilience Plan (NRRP), may represent an opportunity to leverage specific funds for mental health in general, and for youth in particular. Finally, mental health service governance should be harmonized at both national and regional EU levels—with the adoption of best practices implemented by other Member States. This includes, among others, health information system and data collection, which is critical for analysing epidemiological trends and for monitoring and evaluating services, to offer a public and integrated system for the care and protection of young people, in line with the Convention on the Rights of the Child.

## 1. Introduction

Youth is a period of major physical and psychological development as well as modifications in social relations [1], and it is generally considered the healthiest period of life. However, among young people, the disability and mortality burden due to mental health conditions (i.e., mental disorders [MDs]), substance-use disorders (SUDs), interpersonal violence and self-harm are rising [2]; indeed, half of these conditions diagnosed in adulthood emerge by the age of 14 and 75% by the age of 24 [3]. Recent decades have been characterized by a slight but regular increasing burden of MDs in young people, with a concerning 32% of increased burden from 1990 of years of life lost due to eating disorders, and with MDs being the leading cause of disability in people aged 10–24 years old in all European Union Member States in 2019 [4]. These numbers have been estimated to further increase, affected, among others, by the COVID-19 pandemic.

In this light, the WHO European Framework for Action on Mental Health (EFAMH) 2021–2025 [5] provides endeavors to protect mental well-being as an integral element of COVID-19 response and recovery, to fight the stigma and discrimination associated with MDs, as well as to promote investment in accessible quality mental health services.

It is demonstrated that mental health issues are not only related to COVID-19, especially in Italy. In particular, a recent study [6] conducted in central and southern Italy to analyse several 21st-century stressors (e.g., Climate Change, Ukrainian War, COVID-19) and global mental health problems revealed different relationships between 21st-century concerns and depression, anxiety, and stress were observed across age groups. In particular, young adults would seem to be the most concerned group in all categories, while older adults are the least concerned.

Additional longitudinal studies about the impact of COVID-19 on adolescents and young adult Italian population suggest that the pandemic’s persistence over time may have had an impact on youth’s psychopathology, influencing the frequency, type, and complexity of mental health problems [7,8]. Furthermore, evidence revealed that the new generation of adolescents may express their discomfort through alternative-type symptoms rather than the expected “traditional” internalizing forms, including a higher problematic social media usage than peers before the pandemic [9].

The aim of this Short Communication is to investigate the appropriateness of implementing gender- and age-sensitive, integrated, youth-centred mental health services in Italy.

## 2. Materials and Methods

We conducted a systematic search for programmatic and policy documents and reports with a solid literature and policy analysis. The Comment focuses on the weaknesses of youth mental health services, especially underlining the lack of supply where preventive intervention would be most effective. Moreover, programmatic documents from Italian Government have been consulted to summarize the appropriateness of the investment in youth to have benefits during youth itself, in adulthood, and for the next generation, as well as the missed opportunity to use part of the funds of the National Recovery and Resilience Plan (NRRP) for youth mental health. The authors further examined contributions from previous studies of knowledge utilization in mental-health policy-making, including accompanying institutional arrangements. The results take many forms, ranging from national health policies made by the government to clinical guidelines/evidence published in peer-reviewed journals. In terms of the utilization of the knowledge, research-informed policies can be referred to as the primary outputs of this Communication. 

To improve the strength of the comment, and in consideration of the problematic nature of the transition between child mental health services and adult mental health services, primarily based on an age criterion, and the importance of creating ad hoc adolescents and youth care pathways and services, the authors have analyzed the different strategic policy documents on mental health at the regional level.

## 3. Results

### Addressing Mental Health Gaps among Young People in a Post-COVID-19 Era

The burden of MDs, SUDs and self-harm behaviours is a major concern among young people in high-income countries, with common MDs (i.e., depression and anxiety) ranking among the top ten causes of years lived with disability (YLDs) [10]. A recent study from European countries found that 20% of young aged from 10 to 24 years suffered by mental conditions in 2019 [11], with MDs being the leading cause of YLDs in all these countries, and with common MDs ranking between the first and the fourth position in each country. Nonetheless, child and adolescent mental health services (CAMHS) show a high heterogeneity in Europe, and official national adolescent mental health policies have been implemented in only 70% of European countries [12]. The increasing prevalence of mental conditions in young people observed across several European countries requires a deeper understanding of the disease burden associated with these disorders to inform future health policy. This is also striking in relation to the COVID-19 pandemic, which is associated with an increase in mental distress and may result in a rise of mental conditions over the next years [13], as highlighted by the preliminary evidence [14]. According to a survey on 5008 people conducted through the project portal #PRESTOinsieme, during the spring 2020 lockdown, almost 90% of youth over the age of 16 suffered from psychological stress and almost 50% had symptoms of depression [15].

Indeed, the Italian Minister of Health, to support local health authorities, signed the decree n. 15/2022, assigning a bonus to be spent for the psychology services, accounting for an initial 10 M EUR budget, and recently renewed to 25 M EUR. The contributions, up to a maximum amount of EUR 600 per person and granted according to the beneficiaries’ financial condition, represent a first step to address the severe distress caused by the recent COVID-19 pandemic in the population. The initiative showed the specific need to address mental health services in the general population, as in two days 85 thousand applications were received. The initial available budget, however, was sufficient to cover approximately 16 thousand users’ requests [16].

Following decree n. 15/2022, several sporadic conjunctural financing initiatives were also promoted at the regional level, aiming to treat and prevent MDs that emerged with the pandemic. Indeed, Lazio was the first region in Italy to activate “Aiutamente Giovani”, a specific 10.9 M EUR intervention prioritizing school-age adolescents (up to 21 years), supporting youth in coping with post-COVID-19 mental health distress [17]. In Friuli Venezia Giulia, the Regional Government activated the Civil Protection’s toll-free number 800-500-300, reserved for the Coronavirus emergency, which also offers psychological support. From 12 noon to 7 p.m., seven days a week, a professional psychological consultation is offered to people in need. In addition, the region has allocated 441,000 EUR to hiring community psychologists dedicated to youth and adolescents [18]. Among others, Lombardy Region has proposed the “primary care psychologist”, a psychological service provided free of charge to all the residents (12 M EUR resources) for the three-year period 2023–2025. In this regard, the National Council of Psychologists and the Ministry of Education signed a pilot program involving 6000 schools (out of 8000) providing free of charge support to students, staff, and families [19]. The revolution in the mental health system for youth is also impacting Southern Regions. For instance, in 2022, Sicily had a record number in compulsory treatment admissions (857, more than double the national average), and routinely manages 69,000 patients with mental disorders (the rate is 66 new patients for every 10,000 inhabitants, against the Italian average of 50). The current regional youth mental health plan is transitioning towards community mental health networking. In particular, this implies a shift from fragmented and competing local services (and other social agencies) to an integrated mental health department implemented at the community level, in a network with all the other territorial services. Indeed, this is in line with the national policy, which foresees the structuration of mental health department—strongly community-based—in the last 20 years. Additionally, the plan has replaced the payment of the fee-per-bed model for the financing of the individualized therapeutic project through the public health budget and mixed management. There are also pilot initiatives for refugees and asylum-seekers, as the island is constantly at the front line of the migration crisis [20].

Additionally, the 5th Italian National Plan of Action and Interventions for the Protection of the Rights and Development of People in Developmental Age 2022–2023 has recently mentioned the need for school psychology services to target adolescents to provide a common and coherent response to their needs [21], highlighting the lack at primary care level in human resources, as well as a lack of hospital beds for children suffering from neuropsychiatric disorders. According to a recent survey, 81% of Italians declared that they were in favor of establishing school psychologists and identified the most important activities: the listening and support (54%), prevention of discomfort (41%), support for families (29%), advice for the school system as a whole, and support for teachers (18%). Among students (15–18 years old), seven out of ten chose ‘listening and support’, thus highlighting the need to communicate [22].

These aspects have also been highlighted in the Italian National Action Plan for Mental Health (NAPMH) [23], which includes specific objectives and actions to implement care pathways and develop the organization and integration of services, and criteria and indicators for monitoring and evaluation (26 key performance indicators across 8 objectives set by the NAPMH). Overall, the plan offers specificity of care for childhood and adolescent neuropsychic disorders and interventions in the developmental age. For instance, it established standardized multidisciplinary team interventions based on different care intensities according to the disorders, contexts, and developmental stages, and not limited to the complexity and severity of disease. Despite the presence of a dedicated national plan, less than half of the NAPMH objectives have been implemented by Italian Regions. Indeed, according to a recent assessment conducted at the national level [24], the objectives mostly affected by the low implementation rates of NAPMH guidelines are, among others: i. promotion of the physical health of psychiatric patients; ii. diagnosis and treatment of people with age-related mental disorders; iii. prevention and combating stigma.

Additionally, the NAPMH reports a highly fragmented situation in the Child and Adolescent Neuropsychiatry services, which, in some cases, reveals clear shortcomings, particularly in terms of psychiatric acute care, as well as complex and developmental disabilities. The available data indicate an insufficient differentiation of demand, generating an inappropriate use of resources according to the case complexity (e.g., users with severe disorders receiving similar care pathways compared to users with common disorders and vice versa).

The overall strategy of the NAPMH can also be used to derive the “Guidelines for neuropsychological disorders of the developmental age” [25], which aims to provide operational guidelines on shortage of beds in child and adolescent neuropsychiatry services, which jeopardize care pathways and determine the risk of chronicization of disorders. The document highlights the need to implement an integrated and comprehensive regional network of services for the diagnosis, treatment, and rehabilitation of neuropsychological disorders in young people, and foster the development of mental health services that are capable of dealing with the transition from childhood to adulthood. Indeed, transition still represents a major concern, with, in about two-thirds of cases, no specific adult services provided to ensure adequate health responses for young people, resulting in adult psychiatric services only being offered to a fraction of eligible users [21,22]. The underestimation of the transition from child to adult mental health services, from a life-course perspective, may result in a significant drop out/discontinuity of treatment and care, and is associated with high socio-economic costs, including a considerable overload on informal caregivers (Figure 1).

## 4. Discussion

There is an urgent need to address the mental health gaps among young people in a post-COVID-19 era, particularly to arrest the rise in dramatic differences and inequities in terms of health, education, and inclusion opportunities, which impact young people and their future lives, hampering the possibility of the development of entire communities. Wide-ranging interventions, systemic actions, public and integrated services should be implemented and funded, systematically and with a long-term vision, rather than sporadically, and should be included in an overall structural strengthening of the healthcare system. Youth is a period in which investment reaps benefits during youth itself, in adulthood, and for the next generation [26]. In this context, an investment and implementation opportunity was recently missed within the National Recovery and Resilience Plan (NRRP). The NRRP highlights the need to strengthen primary care, but it does not foresee specific funds for mental health in general and for youth in particular [27]. The appropriateness of initial assessments and referrals should be optimized within a multidimensional and integrated approach to the socio-health, socio-educational and justice areas, with a specific focus on the inclusion of all young people, particularly the most vulnerable. Services should be gender- and age-sensitive and responsive, endorsing a youth-centred care, particularly in consideration of the transition from adolescent to adult services. In this context, Veneto Region has been proposing the establishment, on an experimental basis, of a specific Adolescent Neuropsychiatry (NPA), an innovative unit composed by a multiprofessional team, integrated in the Department of Mental Health, based on Individualized Therapeutic Projects, following a youth-centered care approach, which is accessible and attractive to young people [28]. The transition from children to adults’ services should consider organizational dimensions, ensure continuity of care, reduce fragmentation of services, involve the adolescent in the decisional process, supporting his/her autonomy and empowerment. In this context, it is important to plan the transition well in advance, ensuring the continuity of the service and the flexibility in the transition to adulthood with respect to actual age. At the European level, a recent mapping survey was conducted across all 28 Member States, reporting that between 25 and 49% of Child and Adolescent Mental Health Services (CAMHS) users will need to transition to Adult Mental Health Services (AMHS). Estimates of the percentage of AMHS users aged under 30 years who had has previous contact with CAMHS were most commonly in the European Region 20–30% (33% on average). In particular, written policies for managing the interface were available in only four countries, and half (14/28), indicated that no transition support services were available [29]. For example, in Belgium, the 2015 reform of mental healthcare supported the inclusion of youth until the age of 23 in child and adolescent mental health services, and the development of integrated care and community [30]. Moreover, local procedures for integrated transition management should be formalized and shared to ensure that the criteria for accessing adult services are consistent with the needs and expectations of young people, and adequate monitoring should be formally structured. Denmark reported offering transition planning for service users with eating disorders aged 13 years and over. Very interesting findings have also been made available through the MILESTON project, aiming to improve transitions for young people from CAMHS to AMHS in Europe [31]. In this light, differences in service provision can be explained by differences in service configuration. Among others, regional differences in AMHS configuration have been reported, for example, regarding the legal transition age in health care (mostly 18 years, with exceptions: Malta (16 years), Cyprus (15–19 years), Czech Republic (18–19 years), Denmark (17 years), Estonia (19 years), Finland (18–23) and the UK, France (16–18) and the Netherlands (18–21). At EU level, only Cyprus, Denmark, Spain and the UK have written national or regional policies or guidelines to manage the interface between the services. In terms of budgetary and fiscal aid for transition support services, the project indicated the availability of a separate funding system, accessible to CAMHS and AMHS in 10/26 countries (38%), and flexible funding in 5/26 countries (19%).

## 5. Conclusions

The increasing prevalence of mental conditions in young people observed across several European countries, requires a deeper understanding of the disease burden associated with these disorders and a greater commitment to developing youth-centered care mental health services. This is also striking in relation to the COVID-19 pandemic, which is associated with an increase in mental distress and may result in a rise in mental conditions in future, as highlighted by preliminary evidence.

Mental health service governance should be harmonized at national and regional EU level, as well as data collection, which is critical for analysing epidemiological trends and monitoring and evaluating services, to offer a public and integrated system for the care and protection of young people, in line with the Convention on the Rights of the Child. In particular, an appropriate and timely reconfiguration of mental health services is critical to overcome transition-related discontinuity of care; this includes, among others, a joint CAMHS and AMHS review and an adaptation of already existing transition guidelines/best practices to ensure they are fit for country-specific contexts. The allocation of dedicated resources at both national and regional levels needs to be focused on allowing policy-makers and service-managers to access relevant information to evaluate the problem in detail [32].

## Figures and Tables

**Figure 1 ijerph-19-14937-f001:**
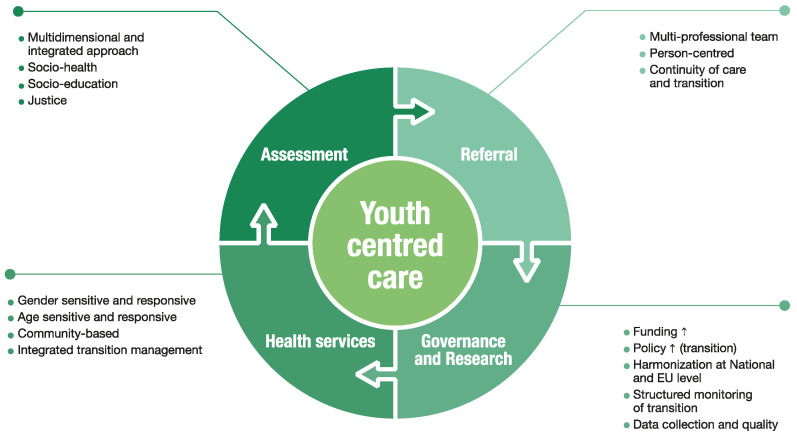
Youth-centred care framework for the structural strengthening of the healthcase system. Design of mental health services across adolescence and the transition to adulthood represents a blueprint for an overdue global system reform. While resources will vary across settings, the mental health needs of young people are largely universal and underpin a set of fundamental principles and design features. These include the introduction of a soft entry point for youth mental health platform, where young people are part of a co-creative development process.

## Data Availability

Not applicable.

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
