# Peer review of "What Is Next for Public Health after COVID-19 in Italy? Adopting a Youth-Centred Care Approach in Mental Health Services"

_ijerph, 2022, doi:10.3390/ijerph192214937_

Round 1
Reviewer 1 Report
This is a very interesting and important paper. The abstract introduces the content of the paper appropriately. The introduced theories and policies are in line with the content and direction of the paper. The aims highlighted in the Materials and methods section is clear and undéstandable. The results introduce theories concerning Italy and the EU which is very important. The differences between the practices applied in the EU should be highlighted in the Discussion section. Here, some detailed practices (not only concerning the EU but some exact countries) can be also mentioned (notifying the room for improvement). Figure 1 is very valuable, but it is unnecessary to put it into a separate subsection, it should be the part of 3.1. Also, the authors can extend its introduction.
Overall, the Discussion section is clear. The Conclusion section should be extended a bit with the result of the comparison of the Italian and international practice, highlighting and suggesting a direction (even with exact examples).
Author Response
Authors would like to thank the reviewer for the positive assessment and the important inputs provided, which helped to significantly improve the quality of the manuscript.
In details, additional elements regarding the differences between the practices applied in the EU have been included in the Discussion section, with Country-specific information mentioned. These elements are also represented in the Conclusion section and include, among others, a direction based on best practices.
Following the reviewer comment, the Figure 1 is now part of 3.1 and it is present extended introduction.
Reviewer 2 Report
Thank you for highlighting this important issue. Indeed, more data, evidence and actions should be taken. COVID-19 pandemic should be use as a stimulus.
Even though I recognize that this is intended to be a COMMUNICATION, I suggest to better highlight:
1. The methodology you used for pooling/collating the evidence. Which method do you use? Please share the information you have, behind and beyond your already published own research (references 2 and 4).
2. In the discussion: can you please highlight some positive examples of the desired model of services? Are there any countries/communities that had used such models before the pandemic? Any difference in the outcome? Can you please provide more concrete guidance on what should be done at the governance, service delivery, resources, etc?
3. You mention Figure 1 in line 159, but the heading of the figure says: 3.2. Please, review and correct. Also, I am not sure if that figure is shedding more light on the issue. Please, review and change.
Author Response
Authors would like to thank the reviewer for the positive assessment and the important inputs provided, which helped to significantly improve the quality of the manuscript.
In details, additional information regarding the methodology used for pooling/collecting the evidence has been provided.
In the discussion, additional elements regarding the differences between the practices applied in the EU have been included in the Discussion section, with Country-specific best practices. More concrete guidance on what should be done at the governance, service delivery, resources et alia have been included.
Figure 1 is now part of 3.1 and it is present extended introduction.
Reviewer 3 Report
Thank you for allowing me to revise the manuscript "What is next for public health after COVID-19 in Italy? Adopting a youth-centred care approach in mental health services".
I think this is a very interesting article. Discussing this issue is urgent and crucial because from my point of view urgent actions (also at the policy level) are needed.
I agree with the authors' claim that "wide-ranging interventions, systemic actions should be implemented, funded, and included in an overall structural strengthening of the healthcare system".
I have read the manuscript with very interest. I think that this communication could be accepted in this current version. I have only a few suggestions:
- Authors should add longitudinal studies about the impact of COVID-19 on adolescents and young adults Italian population.
- In the results section, the Authors present the program activated in some Italian regions. Did Southern Italy remain uncovered?
Author Response
Authors would like to thank the reviewer for the positive assessment and the important inputs provided, which helped to significantly improve the quality of the manuscript.
Several references to longitudinal studies about the impact of COVID-19 on adolescents and young adults Italian population have been reported in the manuscript.
In the results section, additional evidence from the Southern Italy experience is described, leading to a comprehensive description of case studies from Norther, Central and Southern regions.